# Shift work and long work hours and their association with chronic health conditions: A systematic review of systematic reviews with meta-analyses

**Adovich S. Rivera**[1]*, **Maxwell Akanbi**[1], **Linda C. O'Dwyer**[1,2], **Megan McHugh**[1,3]

**1** Institute for Public Health and Medicine, Feinberg School of Medicine, Northwestern University, Chicago, Illinois, United States of America, **2** Galter Health Sciences Library and Learning Center, Feinberg School of Medicine, Northwestern University, Chicago, Illinois, United States of America, **3** Department of Emergency Medicine, Feinberg School of Medicine, Northwestern University, Chicago, Illinois, United States of America

* adovichrivera2021@u.northwestern.edu

**Data Availability Statement:** All relevant data are within the manuscript and its Supporting Information files.

## Abstract

### Background

Previous reviews have demonstrated that shift work and long work hours are associated with increased risk for chronic conditions. However, these reviews did not comprehensively assessed the body of evidence, and some were not conducted in a systematic manner. A better understanding of the health consequences of shift work and long work hours will aid in creating policy and practice recommendations. This review revisits the epidemiologic evidence on the association of shift work and long work hours with chronic conditions with particular emphasis on assessing the quality of the evidence.

### Methods and findings

We conducted a systematic review of systematic reviews with meta-analyses (SR-MA) that assessed the link between shift work or long work hours and chronic conditions (PROSPERO CRD42019122084). We evaluated the risk of bias of each SR-MA using AMSTAR v2 and assessed the overall evidence for each condition using the GRADE approach. We included 48 reviews covering cancers, cardiovascular diseases, metabolic syndrome and related conditions, pregnancy complications, depression, hypertension, and injuries. On average, only 7 of 16 AMSTAR items were fulfilled. Few SR-MAs had a registered protocol and nearly all failed to conduct a comprehensive search. We found moderate grade evidence linking shift work to breast cancer and long work hours to stroke. We found low grade evidence linking both shift work and long work hours with low to moderate increase in risk for some pregnancy complications and cardiovascular diseases. Low grade evidence also link long work hours and depression.

### Conclusions

Moderate grade evidence suggest that shift work and long work hours increase the risk of breast cancer and stroke, but the evidence is unclear on other chronic conditions. There is a

**Funding:** This project is funded through the Robert Wood Johnson Foundation (https://www.rwjf.org/) (Grant Number 7610) awarded to MM. The funders had no role in study design, data collection and analysis, decision to publish, or preparation of the manuscript.

**Competing interests:** The authors have declared that no competing interests exist.

need for high-quality studies to address this gap. Stakeholders should be made aware of these increased risks, and additional screening and prevention should be considered, particularly for workers susceptible to breast cancer and stroke.

## Introduction

Jobs that require work outside the traditional daytime hours of approximately 8 AM to 6 PM have become ubiquitous across economically developed nations. Some jobs such as those in healthcare, manufacturing, and law enforcement routinely require night time or prolonged shifts. In the United States (US) and European Union (EU), a fifth of employees are shift workers [1]. Additionally, a substantial share of the work force works more than the usual 40 hours per week, with 36.1% of the global workforce clocking in excessive hours (more than 48 hours) per week [1].

Shift work schedules and long work hours give rise to acute and chronic health effects ranging from metabolic syndrome to cancers that arise from shared and interacting biological pathways [2–5]. Shift work disrupts a person's circadian rhythm and the internal processes controlled by this rhythm, such as clock genes for cell proliferation and melatonin secretion. These disruptions promote inflammation and oncogenesis and are immunosuppressive[3–5]. For example, breast cancer among female shift works has been attributed to increasing DNA methylation with increasing exposure to shift work[4]. Long work hours, meanwhile, not only cuts into non-work hours that the body needs for rest and recovery but can also be a form of psychologic stress [2,6], and if chronically exposed, this stress can lead to cardiovascular disease [7]. Non-standard work hours can also induce unhealthy coping behaviors, such as low physical activity and poor diets [2].

Recognizing the potentially harmful effects of shift work and long work hours, many governments have enacted laws and regulations restricting their use [8]. While in almost all countries there are restrictions on the maximum allowed work hours and stipulated compensation for work done in excess of these hours, regulation of shift work is more varied. The EU restricts the numbers of night work hours a person can perform per day [8]. The EU, Japan, and South Korea, also require special health examinations for night shift workers and have imposed shift work prohibitions on pregnant women [8]. The US has imposed few restrictions on non-standard work hours, and the policies that exist typically focus on the effects of non-standard work hours on productivity and safety, such as fatigue and occupational injuries [8].

Prior reviews of epidemiologic evidence have concluded that non-standard work hours are associated with increased risk for breast cancer, metabolic disease, and cardiovascular disease [9–12]. However, these reviews lacked a comprehensive approach for judging the body of evidence, considered only select sources of bias, and some were not conducted in a systematic manner. These weaknesses restrict the ability to confidently quantify the increased risk caused by shift work and long work hours, prevent the calculation of the associated healthcare costs, and therefore, fall short of motivating changes in policy and practice.

In this paper, we report on our systematic review of systematic reviews with meta-analysis (i.e., "umbrella review") on the association of exposure by workers to shift work and long work hours (hereinafter referred to as "non-standard work hours") with chronic or high cost conditions. Given the plethora of reviews on various conditions, we felt that compiling and systematically assessing the evidence would be helpful for clinicians and policy makers as they consider appropriate policies and guidelines on these two common non-standard work hour

set-ups. Results may also be of interest to employees, who potentially bear an increased risk of chronic illness, and employers and insurers, which often bear the healthcare costs associated with chronic illnesses. We also identify any gaps in the available body evidence to help direct future research.

## Materials and methods

### Screening, inclusion and exclusion criteria

We included any study that conducted a systematic review of the literature (as opposed to purposive or undocumented selection of articles) using at least one database to examine how exposure by adult workers to shift work or long work hours affect the risk of having or acquiring a chronic or high-cost condition. We included only those that pooled the results in a meta-analysis since only these studies report quantified risk measures that are necessary for making policy decisions regarding non-standard work hours. The screening was done by two authors (AR, MM) independently and differences were resolved by consensus.

We defined shift work as any work outside the standard daytime work hours of approximately 8 AM to 6 PM, and this includes rotating shift work, fixed nights, and evening work. Long work hours was defined as work with a duration that exceeds 40 hours per week. The chronic conditions included in this review were a combination of the highest cost conditions with at least ten percent prevalence among adults in the US, and conditions with the highest personal health spending in the US, for example, diabetes, ischemic heart disease, and low back and neck pain [13,14].

We excluded reviews that investigated (1) biologic mechanisms behind the effects of non-standard work hours on health, (2) association of non-standard work hours and risk factors of chronic conditions (e.g. smoking, low physical activity), and (3) interventions to mitigate effects of non-standard work hours. We excluded reviews that measured impact of non-standard work hours on biomarkers instead of diagnosis (e.g. blood pressure as a continuous outcome vs hypertension), quality of life, sleep-related measures (e.g. disturbances, length, quality), and other employee outcomes (e.g. absenteeism, productivity, work-related stress). Reviews that only had abstracts available were excluded.

We searched the following electronic bibliographic databases from inception to April 2019: MEDLINE Pubmed, Embase (embase.com), Scopus, CINAHL (Ebsco), Web of Science, PsycINFO (Ebsco), ABI Inform Global (Ebsco), Business Source Premier (Ebsco), and Risk Abstracts (ProQuest). We also looked for grey literature in the following sites: Grey Literature Report, OpenGrey, CADTH, Systematic review repository (AHRQ), Epistomonikos, PROSPERO, Working time conference meetings/abstracts, CDC National Institute for Occupational Safety and Health, and Proquest Dissertations. The search strategy was adapted and developed using similar reviews and iterative searches by the research librarian in our team (LO), and a copy can be found in the S1 File. To capture additional relevant literature, we contacted authors of PROSPERO protocols that were marked as completed or were past the registered completion date to ask about availability of their review. We also hand searched the references of included studies to identify potentially relevant articles.

There were no language, country or date restrictions in the search, but we only included articles written in English, Spanish or French due to logistical factors. For Spanish and French articles, an English summary was produced and used for the screening.

### Data extraction and quality assessment

Three authors (AR, MA, and MM) performed data extraction and quality assessment independently using standardized forms with each paper being assigned to two reviewers. Conflicts

were resolved through discussion. We extracted pooled risk estimates, which were commonly reported as odds ratios (OR) or risk ratios (RR). As most meta-analyses had multiple pooled estimates, we extracted the main pooled results for each outcome of interest and any available dose-response results. We also extracted subgroup analyses based on shift type, design (e.g. cohort only), gender, age group, quality assessment (e.g. high-quality studies only), and race.

To provide a comprehensive and transparent assessment of the evidence, we utilized A MeaSurement Tool to Assess systematic Reviews (AMSTAR) v2 to assess the quality of an individual systematic review with meta-analysis (SR-MA) [15], and the Grades of Recommendation, Assessment, Development and Evaluation (GRADE) approach to judge the overall body evidence for each outcome [16]. For each paper, we counted the number of AMSTAR items that were fulfilled and divided by the total items to get a quality score. GRADE classifies evidence into high, moderate, low, or very low, which reflects the confidence that the studies capture the true effect or association of interest. Since all included SR-MAs pooled results from observational studies, the baseline quality of evidence was "low" and we down- or upgraded the assessment of each outcome following the GRADE criteria which considers risk of bias, imprecision, inconsistency (including heterogeneity), indirectness, and publication bias. For both AMSTAR and GRADE assessments, two authors (AR and MM) conducted independent assessments and resolved differences through consensus. In the Results below, we report quantitative findings of conditions with moderate or low grade evidence and the extracted results of individual studies can be found in the S4 File.

### Deviations from the protocol

The review protocol (S5 File) was registered in PROSPERO (CRD42019122084), and we made three modifications. First, we revised the inclusion criteria to limit our scope to SR-MAs. This allowed us to focus on assessing the evidence on the strength of association between non-standard work hours and each outcome. Second, we modified the search strategy to better capture articles that examined the effects of long work hours. The final modification was the adoption of the GRADE approach to assess the overall quality of the evidence. The PRISMA checklist can be found in the S2 File.

### Results

#### Overview of search results

Of the 2,936 articles identified in the initial search, we included 289 in the full text screen. Forty-eight (48) SR-MAs were ultimately included in the analysis (Fig 1) (See S3 File for list of articles excluded in full text screen). Among those articles, 41 (85%) used shift work as the exposure, and 12 (25%) used long work hours. The articles covered the following outcomes: cancers [17–32] (16, 33%), cardiovascular disease [33–41] (9, 19%), metabolic syndrome, diabetes mellitus, and obesity [42–50] (9, 19%), complications of pregnancy [51–58] (8, 17%), depression [43,45,59] (4, 8%), hypertension [60] (1, 2%), and injuries [61] (1, 2%). Some conditions we identified to be chronic or high cost (e.g. low back and neck pain, and lower respiratory tract infections) had not been a subject of eligible systematic review with meta-analysis.

Nearly all SR-MAs had a pooled estimate for the association between exposure to any shift work and an outcome, using "never exposed to shift work" or "regular day shift" as control. The definition of long work hours varied, with the lower limit usually set at or above 40 hours per week. Most SR-MAs used standard hours (35–40 hours per week) as the control. (Table 1)

On average 7 (SD: 2.37) of the 16 AMSTAR items were fulfilled yielding an average score of 44.7% (SD: 14.8). Only 19 (39.5%) SR-MAs had a score of 50% or higher. Few SR-MAs had a registered protocol and nearly all failed to conduct a comprehensive search. We flagged several

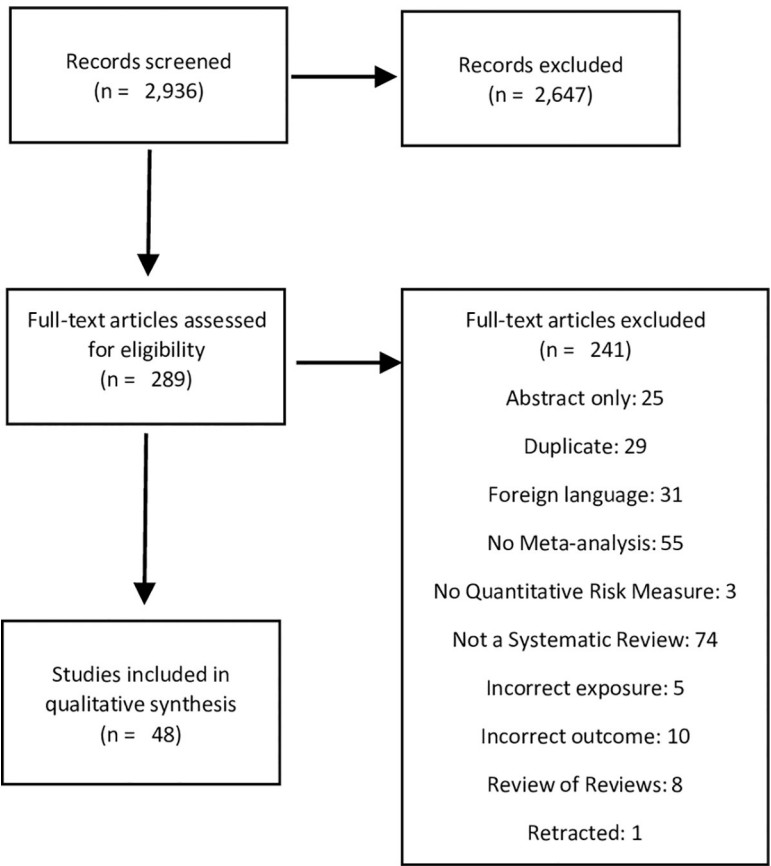

**Fig 1. PRISMA flowchart for systematic reviews.**

SR-MAs for inappropriate analysis due to pooling of hazards ratio with odds and risk ratios. (Fig 2) GRADE assessment of evidence with the pooled risk estimate from the SR-MA with the highest AMSTAR score for each outcome is summarized in Table 2.

## Shift work

We found moderate grade evidence for the association between breast cancer and shift work. Most [18,20,23,26,28,31,32] (7 out of 9) included SR-MAs on breast cancer found a significantly increased risk among shift workers compared to non-shift workers. Focusing on the most recent SR-MAs, which included only high-quality articles, Li (2015) [31] found that shift workers have an 11% increased risk for breast cancer compared to non-shift workers (RR 1.11, 95% CI 1.02 to 1.20, $I^2$ = 48%). Ijaz et al. (2013) [27] also detected a significant dose-response relationship with 5% increase in risk for every five years of exposure (RR 1.05, 95%CI 1.10 to 1.10, I2 = 55%). Lin et al (2015) [37] found a significant association between rotating shift work and breast cancer, although this should be interpreted with caution due to the review's low quality.

Six SR-Mas [33,36–38,40,41] investigated the association between cardiovascular disease and shift work. We found low grade evidence for ischemic heart disease, myocardial infarction, and ischemic stroke. Cheng et al. (2019) [33] found that there was a 13% increase in risk for ischemic heart disease among shift workers versus controls (1.13, 95%CI: 1.08 to 1.20, $I^2$ = 52.7). While they [33] detected a significant dose-response relationship for ischemic heart

**Table 1. Characteristics of included reviews.**

| Author—Year | Chronic condition (s) | Exposure(s) assessed | Control | Types of Studies included | Databases search | Inclusive Search dates | Studies Included in meta-analysis | AMSTAR score |
|---|---|---|---|---|---|---|---|---|
| **CANCERS** | | | | | | | | |
| He 2015[26] | Breast cancer | Any type of shift work | Not explicitly reported | cross-sectional, case-control, cohort | Pubmed | inception to January 2014 | 15 | 4 |
| Ijaz 2013[27] | Breast cancer | Evening, Fixed, Night, Rotating | Day work | case-control, cohort | Pubmed, EMBASE, CINAHL, PsycInfo, LILACS, OSH Update and ProQuest dissertation and theses databas | up to May 2012 | 12 | 14 |
| Jia 2013[28] | Breast cancer | Night shift | Preferred reference was the absence of night work | case-control, cohort | Pubmed, EMBASE, CNKI, Chinese Wanfang Database | 1980 to Sept 2012 | 13 | 8 |
| Kamdar 2013[29] | Breast cancer | Fixed, Night/ Overnight, Rotating | Never had a night shift | case-control, cohort | PubMed, Embase, CINAHL, Proquest Digital Dissertations, and Web of Science (Conference Proceedings Citations Index) | inception to March 1 2012 | 16 | 7 |
| Kolstad 2010[30] | Breast cancer | Night, Rotating, Unspecified | Not clearly stated | case-control, cohort | PubMed, Science Citation Index | inception to May 2007 | 9 | 4 |
| Li 2017[31] | Breast cancer | Night, rotating work that included any number of hours between 000 and 0500 | Day workers | case-control, cohort | Pubmed, Embase | Medline (1946 to 2015 March 10) and Embase (1974 to 2015 March 10) | 20 | 8 |
| Lin 2015[37] | Breast cancer | Fixed, Night, Rotating | none or regular day | Prospective cohort studies | Pubmed, ProQuest | inception to September 2014 | 16 | 5 |
| Megdal 2005[20] | Breast cancer | Any work that included Night/ overnight | No night work or in trades with less than 40% night work | cohort, case-control | PUBMED | January 1960 to January 2005 | 6 | 6 |
| Travis 2016[22] | Breast cancer | Any night, Rotating | never night shift, day work | prospective cohort | Pubmed, Scopus, Web of Science | up to December 31, 2015 | 10 | 8 |
| Wang 2013[23] | Breast cancer | Fixed, Rotating | no exposure | Cohort, nested case-control, case-control | PUBMED, Embase, PSYCInfo, APC Journal Club and Global Health | January 1971 to May 2013 | 10 | 4 |
| Wang 2015[24] | Colorectal cancer | Night shift (ever or regular) | never night shift; regular daytime shift | Cohort, case-control | PubMed, Web of Science, Cochrane Library, EMBASE and the Chinese National Knowledge Infrastructure databases | Inception till March 2015 | 6 | 7 |
| Du 2017[17] | Prostate Cancer | Night, Airline-related, Unspecified | Not specified | prospective or retrospective cohort design | PubMed, ScienceDirect, and Embase (Ovid) | inception to February 4, 2017 | 9 | 8 |
| Gan 2018[25] | Prostate Cancer | Evening, Night, Mixed, Rotating | Not explicitly reported | case-control, cohort | PubMed, Embase, Web of Science and China National Knowledge Infrastructure | up to September 2017 | 15 | 7 |

*(Continued)*

**Table 1.** (Continued)

| Author—Year | Chronic condition (s) | Exposure(s) assessed | Control | Types of Studies included | Databases search | Inclusive Search dates | Studies Included in meta-analysis | AMSTAR score |
|---|---|---|---|---|---|---|---|---|
| Mancio 2018[19] | Prostate cancer | Fixed, Rotating | daytime work | cohort, case-control | Pubmed | inception to 17 November 2016 | 9 | 7 |
| Rao 2015[21] | Prostate cancer | Any type | daytime, fixed day, or never shift work | cross-sectional, cohort, case-control | EMBASE, PubMed, Ovid, Web of Science, the Cochrane register, and the China National Knowledge Infrastructure databases | January 1966 to December 25, 2014 | 8 | 7 |
| Erren 2008[18] | Breast and Prostate Cancer | Night, Rotating, flight attendants | Daytime workers | case-control, cohort | Pubmed, ISI Web of Knowledge | inception to March, 2007 | 7 | 4 |
| Liu 2018[32] | Breast, Digestive System, Hematological system, Prostate, Reproductive system, Lung, Skin cancers | Fixed, Rotating, Mixed | Never or shorter duration night shift | case-control, cohort, nested case-control study | PubMed, Embase, Web of Science | Inception to May 2018 | 58 | 7 |
| **COMPLICATIONS OF PREGNANCY** | | | | | | | | |
| Bonde 2013 [51] | Pregnancy | 3-shift work, Evening/night work, changing shift, work before 0800 or 1800 | Day work, no shift work, all women working >30 hours/week | Cross-sectional, case-control, cohort | Pubmed, EMBASE | Jan 1966 to June 2012 | 13 | 8 |
| Bonzini 2007 [53] | Pregnancy | Fixed, rotating/ changing, or unspecified | Not shift work or day only | Cross-sectional, case-control, cohort | Pubmed, Embase | 1966 to December 2005 | Preterm: 14 LBW: 6 | 10 |
| Bonzini 2011 [52] | Pregnancy | Night, rotating, Unspecified | Working women not exposed to shift work | cross-sectional, case-control, cohort | Pubmed bibliographic databases | 1966 to February 2010 | Preterm: 16 LBW: 6 SGA 10 | 6 |
| Cai 2019 [54] | Pregnancy | rotating, fixed night, long work hours—more than 40 hours per week | fixed day or standard working hours, < = 40 hours per week | cross-sectional, case-control, cohort | MEDLINE, EMBASE, Cochrane Library, CINAHL, ClinicalTrials.gov, Science Citation Index Expanded and Conference Proceedings Citation | up to march 15, 2019 | 62 in SR, 59 in MA | 15 |
| Mozurkewich 2000[55] | Pregnancy | Any type, night, rotating | We considered a subject to be "exposed" if she continued to have the assessed work-related exposure at least through the second trimester of pregnancy. | cross-sectional, case-control, cohort | Pubmed | 1966 to August 1999 | 6 | 7 |
| Palmer 2013[51] | Pregnancy | Any type, evening, fixed, night, rotating, unspecified | daytime work | case-control, cohort, cross-sectional | Pubmed, Embase | 1966 to 31 December 2011 | preterm: 19 SGA: 11 | 6 |
| Quansah 2010 [57] | Pregnancy | Any type | "not exposed" | cohort, case-control, cross-sectional | Pubmed, Embase | January 1966 through August 2009 | 4 | 5 |

(*Continued*)

**Table 1.** (Continued)

| Author—Year | Chronic condition (s) | Exposure(s) assessed | Control | Types of Studies included | Databases search | Inclusive Search dates | Studies Included in meta-analysis | AMSTAR score |
|---|---|---|---|---|---|---|---|---|
| van Melick 2014 [58] | Pregnancy | Any type | no shift work, 40 hours per week | cohort, case-control | Pubmed, Embase | 1990 to Nov 1 2013 | 11 | 8 |
| **CARDIOVASCULAR, METABOLIC AND OTHER CONDITIONS** | | | | | | | | |
| Cheng 2019[33] | CVD | Night work, rotating, irregular/other, mixed following International Labor Organization standards | Daytime workers | case–control or cohort study | PubMed, Web of Science and Embase | January 1970 to October 2017 | 21 | 8 |
| Kang 2012[34] | CVD | Long work hours: >40 hours per week (lower limit varies but 40 seems to be the lowest) | lowest category levels of working hours in each of the studies e.g. if reported <40, 40 to 50, 50 to 60, and > = 60, they used < 40 | case–control study or cohort study | MEDLINER (PubMed), EMBASE, and the Cochrane Central Register of Controlled Trials | Up to March 2011 to September 2011. | 11 | 6 |
| Kivimaki 2015 [35] | CVD | Long work hours: Published studies: varied from 45 h or more to 47 to 55 h or more per week. Unpublished data: > = 55 h per week | Published studies: standard working hours. Unpublished data: 35 to 40 hours per week | cohort | Embase, Pubmed, Individual-Participant-Data Meta-analysis in Working Populations (IPD-Work) Consortium | inception to Aug 20, 2014 | CHD: 5 published, 20 unpublished Stroke: 1 published, 16 unpublished | 11 |
| Li 2016[36] | CVD | Evening, Irregular, Mixed, Night, Rotating, Unspecified shifts | No shift work | cohort (prospective, retrospective cohort, nested case–control | PubMed, Embase, and ISI Web of Science databases | up to 22 December 2015 | 5 | 4 |
| Torquati 2018 [38] | CVD | Fixed, rotating, mixed, any work that differed from standard hours (07:00/08:00–17:00/18:00) | non-shift workers (ie, those who only worked usual daytime hours, 08:00–17:00 hours) | case-control or cohort | Pubmed, Scopus, Web of Science | 2006 to 2016 | 21 | 9 |
| Virtanen 2012 [39] | CVD | long working hours: from ≥10 to >11 h per day or > 40 to 60 h per week | those who worked "normal" hours | cross-sectional, cohort, case-control | Medline | inception of the database (1966) until January 19, 2011 | 12 | 5 |
| Vyas 2012[40] | CVD | Evening, Mixed, Night, Rotating, Unspecified/Irregular | Most studies (n = 30) used non-shift day workers as the referent category, and the remainder used the general population as controls (n = 4). | cohort (prospective and retrospective), case-control | Pubmed including PrePubmed, Embase, BIOSIS Previews, Cochrane CENTRAL, Conference Proceedings Citation Index-Science, Google Scholar, ProQuest Dissertation Abstracts, Scopus, and Science Citation Index Expanded | inception until 1 January 2012 | 34 | 8 |

*(Continued)*

**Table 1.** (Continued)

| Author—Year | Chronic condition (s) | Exposure(s) assessed | Control | Types of Studies included | Databases search | Inclusive Search dates | Studies Included in meta-analysis | AMSTAR score |
|---|---|---|---|---|---|---|---|---|
| Wang 2018[41] | CVD | Rotating, Mixed, Unspecified/ Irregular | non-shift day workers | Cohort studies | Pubmed, Embase | inception to 1 December 2017 | 5 | 7 |
| Angerer 2017 [59] | Depression | night shift work: shift work that included night work between 11 p.m. and 6 a.m | Working during the day or with a varying frequency of night shifts | Longitudinal studies: cohort study, case-control study, quasi-experimental study | PubMed, Scopus, PsycINFO, PSYNDEX, Medpilot. | Start of database to Oct 2015 | 5 | 10 |
| Lee 2017[62] | Depression | Night shift | not specified | cross-sectional, longitudinal, cohort | Pubmed, Embase | PubMed (1970 to August 2016) and EMBASE (1987 to August 2016) | 11 | 5 |
| Virtanen 2018 [63] | Depression | most often defined as ≥55 weekly hours | shorter hours (usu standard hours (most often 35–40 hours)) | large prospective studies including cohort studies with both published and unpublished data. | PubMed and Embase, Web of Science | up to January 2017 | published: 10, unpublished: 18 | 11 |
| Watanabe 2016 [64] | Depression | Long work hours: beyond normal (35–40) hours per week | Normal work hours | prospective cohort (including nested case-control) | MEDLINE (PubMed), PsycINFO, and PsycARTICLES | search done on 15 July 2016 | 7 | 9 |
| Anothaisintawee 2016 [42] | Diabetes mellitus | Rotating shift work, Unspecified shift work. | Regular day workers | Cohort studies | Pubmed, Scopus | Inception through November 2013 | 9 | 10 |
| Cosgrove 2012 [43] | Diabetes Mellitus | Long work hours (>50 h overtime per month, or > = 11 hours per day, or > = 61 hrs per week | 0 to 25 h of overtime per month or <8h/day or 21–40 h per week | cross-sectional, cohort, case-control. for long work hours, all studies were cohort | Pubmed, Allied and Comp Med, British Nursing Index 1994, Kings Fund, CINAHL, DH data, EMBASE, PsychInfo from 1806, major diabetes journals (Diabetes, Diabetes Care, Diabetologia, Diabetic Medicine, Diabetes Research and Clinical Practice, Diabetes Metabolism Research and Reviews) from | 1806 to March 2010 | 3 | 6 |
| Gan 2015[44] | Diabetes mellitus | Evening, Irregular, Night, Rotating, Mixed, Unspecified | Not explicitly reported | cross-sectional, case-control, cohort | PubMed, Embase, Web of Science, ProQuest Dissertation and Theses | up to April 2014 | 12 | 6 |
| Kivimaki 2015 [45] | Diabetes mellitus | long working hours as 55 h or more of work per week | the reference category as 35–40 h of work per week | prospective cohort | PubMed, Embase | up to April 30, 2014 | 4 studies + 19 datasets | 4 |

(*Continued*)

**Table 1.** (Continued)

| Author—Year | Chronic condition (s) | Exposure(s) assessed | Control | Types of Studies included | Databases search | Inclusive Search dates | Studies Included in meta-analysis | AMSTAR score |
|---|---|---|---|---|---|---|---|---|
| Manohar 2017 [60] | Hypertension | Rotating | Individuals with non-shift work status | cohort, cross-sectional or case-control | Ovid PUBMED, EMBASE, Cochrane Database of Systematic Reviews, Cochrane Central Register of Controlled Trials | inception to October 2016 | 27 | 8 |
| Watanabe 2018 [50] | Metabolic Syndrome | Night, Rotating, Unspecified | daytime or not shift work | prospective cohort | PubMed, Embase, PsycINFO, PsycARTICLES and the Japan Medical Abstracts Society databases | up to 2016 | 3 | 7 |
| Wang 2014[49] | Metabolic Syndrome/ Obesity | Fixed or rotating based on International Labor Organization definition | unclear | cohort, case-control, cross-sectional | PubMed and Embase | 1971 to 2013 | 13 | 6 |
| Liu 2018[46] | Obesity | Rotating, Night, Mixed | non-shift workers (8-hour day shift workers) | cohort, cross-sectional, case-control | Pubmed, Embase | inception to December 2017 | 23 | 4 |
| Saulle 2018[47] | Obesity | Any type | day shift | cross-sectional, cohort | Pubmed, Scopus | search done on May 2016 | 4 | 5 |
| Sun 2018[48] | Obesity | Fixed, night, rotating | not reported | cross-sectional, cohort | Pubmed | 42795 | 28 | 7 |
| Fischer 2017[61] | Occupational Injury | Afternoon or evening, Night or graveyard | Morning or day shift | case-control, cross-sectional, and retrospective and prospective cohort studies | Pubmed | up to April 4, 2016 | 29 | 8 |

CVD–cardiovascular disease, SGA–small for gestational age, LBW–low birth weight

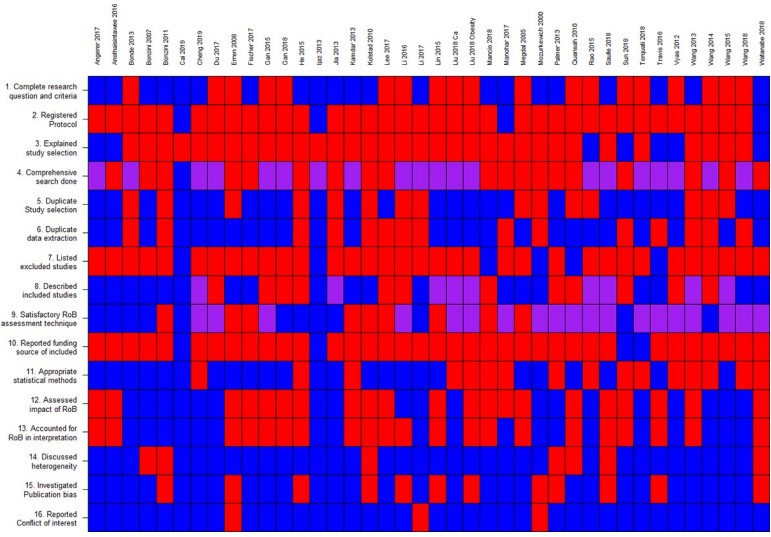

**Fig 2. Quality assessment of included reviews using AMSTAR v2 (n = 41).** Red is not fulfilled, blue is fulfilled, and purple is partially fulfilled. RoB means risk of bias.

disease such that each year of shift work led to a 0.9% increase in risk for ischemic heart disease (RR 1.009, 95% CI: 1.006 to 1.012, $p_{hetertogeneity} > 0.05$), we did not upgrade the evidence rating due to significant heterogeneity of the main and dose-response analyses. This increased risk for ischemic heart disease was present both rotating and fixed night shift work, and they also reported an increased risk for myocardial infarction (1.27, 95%CI: 1.17 to 1.39, $I^2 = 0$) [33]. Meanwhile, Vyas et al. (2012) [40] reported a 5% increase in risk for ischemic stroke (RR 1.05, 95%CI 1.01 to 1.09, $I^2 = 0$) among shift workers. We rated evidence that linked shift work to broadly defined cardiovascular disease (vs specific forms such as stroke) was very low (Table 2).

Among complications of pregnancy, the association between shift work and two complications, preterm delivery and small-for-gestational-age infant was supported by low-grade evidence. Bonzini et al. (2007) [53] estimated a significant increase in risk for preterm birth among shift workers (RR 1.20, 95%CI: 1.01–1.42, $p_{heterogeneity} = 0.002$). They also found that results that trended towards significance on shift work and small-for-gestational-age (RR 1.07, 95%CI: 0.96–1.19, $I^2 = 3.3\%$). The lower quality update of Bonzini et al. 's review [52] found significant increased risk for both outcomes among shift workers.

Cai et al. (2019) [54] estimated the effects of rotating shifts and fixed night shifts on complications of pregnancy and rated the link between the two types of shift work and gestational hypertension to be low grade. Fixed shifts were associated with gestational hypertension (OR = 1.19, 95%CI: 1.1 to 1.29, $I^2 = 0\%$) and there was a trend towards significance between rotating shift and gestational hypertension (OR = 1.19, 95%CI: 0.97 to 1.45, $I^2 = 0\%$).[54] Fixed shifts were also associated with increased risk for preterm delivery (OR = 1.21, 95%CI: 1.03 to 1.42, $I^2 = 36\%$). Rotating shifts, meanwhile, was associated with increased the risk of preterm delivery (OR = 1.13, 95%CI: 1.00 to 1.28, $I^2 = 31\%$), and offspring that are small for gestational age (OR = 1.18, 95%CI: 1.01 to 1.38, $I^2 = 0\%$). We assessed the evidence for miscarriages and preeclampsia to be very low due to imprecision or inconsistency.

We found very low-grade evidence regarding shift work's effect on risk for depression, diabetes mellitus, hypertension, miscarriages, occupational injuries, obesity, metabolic syndrome, and other cancers (colorectal, hematologic, lung, prostate, reproductive system, and skin) (Table 2). Most of the SR-MAs of these outcomes had high risk of bias. The common unmet AMSTAR items are related to protocol registration, search comprehensiveness, listing excluded studies, assessing impact of funding sources, use of appropriate pooling technique, and assessing and discussing risk of bias of included studies in the meta-analysis. (Fig 2) There were also limitations related to imprecision or unexplained heterogeneity.

## Long work hours

Stroke is the only outcome that had moderate grade evidence associated with long work hours. Kivimaki et al. (2015) [35] found that there was a 33% increased risk of stroke among those who worked more than 40 hours per week, compared to those who work standard hours (RR: 1.33, 95%CI: 1.11 to 1.61, $I^2 = 0$). Their metaregression results suggested that risk might be higher among those with high socio-economic status (compared to low socio-economic status) but there were no differences by age group or sex. They also observed a dose-response relationship with longer hours per week translating to higher risk (RR 1.11, 95%CI: 1.05 to 1.17; 11% increase per increase in work hour category).

There was low-grade evidence supporting the association between depression, coronary disease, and selected complications of pregnancy (preterm delivery and low birthweight) with long work hours. Virtanen et al. (2018) [63] found 14% higher odds (OR 1.14, 95%CI 1.03 to 1.25, $I^2 = 45.1\%$) of depression among those who worked >40 hours per week. Kivimaki et al.

**Table 2. GRADE assessment summary of findings.**

| Outcome | Number of SR/ MA | Risk estimate from review with highest quality score[a] | Quality of EVIDENCE | Comments |
|---|---|---|---|---|
| **A) Shift work** | | | | |
| Breast Cancer | 12 | Every exposed: RR 1.1 (1.03 to 1.18, $I^2$ = 62)[31] | ⊕⊕⊕ Moderate | Upgraded due to dose response |
| | | Dose response (every 5 years): RR 1.05 (1.01 to 1.10, $I^2$ = 55)[27][b] | | |
| Ischemic heart disease | 1 | RR 1.13 (1.08 to 1.20, $I^2$ = 52.7) [33] | ⊕⊕ Low | |
| Ischemic stroke | 2 | Risk Ratio 1.05 (1.01 to 1.09, $I^2$ = 0)[40] | ⊕⊕ Low | |
| Gestational Hypertension | 1 | OR 1.19 (0.97 to 1.45, $I^2$ = 2)[54][c] | ⊕⊕ Low | |
| Myocardial infarction | 2 | RR 1.27 (1.17 to 1.39, $I^2$ = 0)[33] | ⊕⊕ Low | |
| Preterm delivery | 5 | RR 1.2 (1.01 to 1.42, $p_{het}$ = 0.002)[53] | ⊕⊕ Low | Significant heterogeneity but robust conclusions in subgroup analysis |
| Small for gestational age | 4 | RR 1.07 (0.96 to 1.96, $p_{het}$ = 0.51)[53] | ⊕⊕ Low | |
| All-cause mortality | 2 | Risk Ratio 1.04 (0.97 to 1.11)[40] | ⊕ Very low | Downgrade for heterogeneity and imprecision |
| CVD: any event (CHD, IHD, MI, stroke) | 2 | ES 1.17 (1.09 to 1.25, $I^2$ = 67)[38] | ⊕ Very low | Downgrade for heterogeneity No upgrade for dose response due to low quality of review |
| Depression | 2 | Risk Estimate 1.42 (0.92 to 2.19, $I^2$ = 74.4)[59] | ⊕ Very low | Downgrade for high risk of bias, heterogeneity, and imprecision |
| Diabetes mellitus | 2 | RR 1.4 (1.18 to 1.66, $I^2$ = 95)[42] | ⊕ Very Low | Downgrade for heterogeneity |
| Hypertension | 1 | OR 1.10 (1.00 to 1.20, $I^2$ = 85)[60] | ⊕ Very Low | Downgrade for heterogeneity |
| Low birth weight | 2 | OR 1.27 (0.93 to 1.74, $p_{het}$ = 0.39)[52] | ⊕ Very Low | Downgrade for high risk of bias and imprecision |
| Metabolic Syndrome | 2 | RR 1.59 (1.00 to 2.54, $p_{het}$ = 0.049)[50] | ⊕ Very low | Downgrade for high risk of bias and publication bias |
| Miscarriage | 3 | OR 1.12 (0.96 to 1.3, $p_{het}$ = 0.53)[51] | ⊕ Very low | Downgrade for high risk of bias |
| Obesity | 4 | OR 1.25 (1.11 to 1.41, $I^2$ = 95.9)[48] | ⊕ Very low | Downgrade for high risk of bias and heterogeneity |
| Occupational Injuries | 1 | RR 1.33 (0.98 to 1.8, $I^2$ = 98.4)[61][b] | ⊕ Very low | Downgrade for high risk of bias, heterogeneity, and imprecision |
| Preeclampsia | 1 | OR 1.05 (0.63 to 1.75, $I^2$ = 0%)[54][d] | ⊕ Very low | Downgrade for imprecision |
| Colorectal cancer | 2 | OR 1.15 (1.01 to 1.32, $I^2$ = 40.2)[32] | ⊕ Very low | Downgrade for high risk of bias and publication bias |
| Hematologic cancers | 1 | OR 1.08 (0.99 to 1.17, $I^2$ = 54.7)[32] | ⊕ Very low | Downgrade for high risk of bias |
| Lung cancer | 1 | OR 1.08 (0.87 to 1.35, $I^2$ = 53.4)[32] | ⊕ Very low | Downgrade for high risk of bias |
| Prostate cancer | 5 | 1.05 (1.00 to 1.11, $I^2$ = 24)[17] | ⊕ Very low | Downgrade for publication bias No upgrade for dose-response due to low quality of SR |
| Reproductive system cancers | 1 | OR 1.06 (0.85 to 1.32, $I^2$ = 49.5)[32] | ⊕ Very low | Downgrade for high risk of bias |
| Skin cancer | 1 | OR 0.93 (0.5 to 1.74, $I^2$ = 74.9)[32] | ⊕ Very low | Downgrade for high risk of bias and publication bias |
| **B) Long work hours** | | | | |
| Stroke | 1 | RR 1.33 (1.11 to 1.61, $I^2$ = 0)[35] | ⊕ Moderate | Upgrade due to dose response |
| Coronary disease | 2 | RR 1.13 (1.02 to 1.26 $I^2$ = 0)[35] | ⊕⊕ Low | |
| Depression | 2 | OR 1.14 (1.03 to 1.25, $I^2$ = 45.1)[63] | ⊕⊕ Low | |
| Low birthweight | 1 | OR 1.43 (1.11 to 1.84, $I^2$ = 0)[54] | ⊕⊕ Low | |
| Preterm delivery | 2 | OR 1.12 (1.11 to 1.33, $I^2$ = 30)[54] | ⊕⊕ Low | |
| Any CVD (CHD, IHD, MI) | 1 | OR 1.37 (1.11 to 1.70, $p_{het}$ = 0.037)[34] | ⊕ Very low | Downgrade evidence for high risk of bias and heterogeneity |
| Diabetes Mellitus | 2 | RR 1.14 (0.35 to 3.72, $I^2$ = 67)[43] | ⊕ Very low | Downgrade for high risk of bias, heterogeneity, imprecision, and publication bias |
| Gestational Hypertension | 1 | OR 0.99 (0.72 to 1.37, $I^2$ = 62)[54][c] | ⊕ Very low | Downgrade for heterogeneity |
| Miscarriage | 2 | OR 1.36 (1.25 to 1.49, $p_{het}$ = 0.02)[51] | ⊕ Very low | Downgrade for heterogeneity and publication bias |

*(Continued)*

**Table 2.** (Continued)

| Outcome | Number of SR/ MA | Risk estimate from review with highest quality score[a] | Quality of EVIDENCE | Comments |
|---|---|---|---|---|
| Small for Gestational Age | 1 | OR 1.16 (1.0 to 1.36, $I^2$ = 57)[54] | ⊕ Very Low | Downgrade for heterogeneity |

[a]–exposure is any type of shift work or >8 hours work per day unless specified otherwise

[b]–increase in risk every 5 year increase in exposure to shift work

[c]—exposure is rotating shift work

[d]–exposure is fixed shift work, GRADE assessment: ⊕ - very low, ⊕⊕ - low, ⊕⊕⊕ - moderate, $p_{het}$–p-value for heterogeneity test

(2015) [54] found a 13% increased risk for coronary disease for those doing long work hours (RR 1.13, 95%CI: 1.02 to 1.26 $I^2$ = 0).[35] Cai et al. (2019) found a 21% increase in odds for preterm delivery (OR 1.21, 95%CI: 1.11 to 1.13, I2 = 30%) and a 43% increase in odds of having low birth weight offspring (OR 1.43, 95%CI: 1.11 to 1.84, I2 = 0%) among women working >40 hours per week during pregnancy. They also reported a significant linear relationship between hours worked and the risk of preterm delivery.

We found very low-grade evidence for long work hours and the following outcomes: miscarriage, preeclampsia, gestational hypertension, small for gestational age, diabetes mellitus, and any cardiovascular disease (Table 2). SR-MAs had high risk of bias and issues with the heterogeneity of results casting doubt on the significant relationships detected by these reviews. Unmet AMSTAR items leading to the high risk of bias were similar to that of reviews on shift work.

## Discussion

We found moderate grade evidence linking shift work to breast cancer and long work hours to stroke. We also found low-grade evidence linking shift work to ischemic heart disease, myocardial infarction, ischemic stroke, gestational hypertension, preterm delivery, and small-for-age babies, and low-grade evidence on the association between long work hours and coronary disease, depression, low birthweight babies, and preterm delivery (Table 2). Our conclusions align with previous reviews[10,11], though we identified specific diseases (e.g. stroke and myocardial infarction) rather than adopting broad categories (e.g., cardiovascular disease). Our study is notable because we found that not all associations of non-standard work hours to cardiovascular diseases and cancers were supported by sufficient quality evidence.

Our findings should be of interest to workers, unions, and other organizations that advocate for workplace well-being. Workers should be informed of the risks associated with these jobs and the evidence-based screenings and interventions that might mitigate the risk. The increased occurrence of these outcomes ultimately translates to increased healthcare costs, which burdens the workers and businesses. We recognize that in some industries (e.g., manufacturing, transportation, hospitals, and police/fire departments), shift work and long work hours may be inevitable; nevertheless, employers should be aware of the healthcare costs associated with their non-standard work hours, potentially consider alternative schedules, and encourage screenings and interventions to reduce risk. In countries like the US, where self-insured companies bear the additional costs of these conditions, there might be a case for restricting non-standard work hours that balances lost productivity with potential savings due to the prevention of these conditions.

Findings may also be of interest to policymakers, as several countries have enacted policies to protect shift workers such as restricting total shift work hours per week and requiring companies to offer free health exams to shift workers [8]. It is still rare for governments to require

companies to provide compensation to employees harmed by shift work. The exception is Denmark, where shift workers who develop breast cancer receive compensation. The first claimants received amounts ranging from US$ 3,000 to US$ 100,000, funded through their employers' insurance companies [65]. However, current evidence does not clearly identify advantages of fixed shifts over rotating shifts and does not suggest a specific threshold for maximum number of hours per week. Both topics warrant more research. Our findings, however, suggest that a maximum lifetime exposure cap might be warranted, particularly when the risk from exposure has meaningful dose-response association. This policy should be considered for occupations where shift work is unavoidable, such as healthcare and law enforcement. In countries with universal healthcare insurance, regulating non-standard work hours may translate into non-trivial societal savings.

Screening and behavioral changes are commonly used preventive interventions for various health conditions. While we found moderate evidence suggesting non-standard work hours may increase the risk for certain conditions captured in current screening or preventive guidelines (e.g. breast cancer and stroke); there are no specific recommendations for screening for these conditions based on work hours. It is unclear whether there should be unique guidelines for those exposed to non-standard work hours (e.g. should shift workers be screened at earlier ages?). Some research exists on behavioral interventions for shift workers [66]. Guidance for employers on managing the impact of shift work remains largely focused on designing efficient shift schedules and promoting healthy lifestyle among workers [66,67]. The effect of these interventions in the long term and on the risk for acquiring chronic conditions should continue to be investigated.

We performed a comprehensive search and assessment of the literature to arrive at our findings. We used a reproducible method for assessing the evidence, and as new reviews are produced, our assessments can be updated following the same methodology. Despite the number of SR-MAs included, several research questions related to the epidemiologic link between non-standard work hours and chronic conditions remain unanswered. Studies on differential risks are needed. Examples would be studies that compare rotating versus fixed shift and studies that look at sex or geographic differences. Dose-response meta-analyses are also needed, especially for establishing causality.

We focused on epidemiologic evidence in this review, but it should be acknowledged that mechanistic studies that investigate how shiftwork alters biological processes to increase the risk for various conditions are necessary to prove with certainty that non-standard work hours are causing these outcomes. These studies, together with epidemiologic studies, are also needed to guide intervention and policy development. We found studies that proposed disease mechanisms for two conditions with moderate grade evidence. For breast cancer, shift work leads to disruption of circadian rhythms which in turn lead to genetic and epigenetic changes that promote cancer growth [4]. For stroke, long work hours is a source of stress and this stress leads to damage to the cardiovascular system. Long work hours can also promote unhealthy behaviors that further increase risk for stroke [6,7].

Several SR-MAs included in this study failed to meet the current standards for systematic reviews and meta-analyses as outlined in AMSTAR. Requirements such as protocol registration, comprehensive search strategies, and appropriate pooling of studies were most commonly unmet. Failing to meet AMSTAR conditions was a common reason for downgrading the evidence. We recommend that future SR-MAs are conducted in accordance with these guidelines to ensure minimization of risk of bias.

We downgraded much of the evidence due to issues of heterogeneity. The individual studies pooled by meta-analyses that we reviewed often had differences in definitions, and measurement of exposures, and included populations. There were also differences in the variables

used to calculate adjusted risk estimates. Despite these potential sources of heterogeneity, subgroup analyses often failed to identify any socio-demographic or study design-related factors as a significant source of heterogeneity. Fortunately, individual level cohort data is increasingly becoming available and allows for individual-level meta-analyses. These individual-level meta-analyses allow researchers to apply more consistent definitions and utilize the same regression models for getting adjusted measures of association.

Our review has several limitations. One is that we focused on SR-MAs and did not look for additional observational studies. Some of the SR-MAs we included (e.g., those regarding pregnancy complications) were published more than five years ago, and new observational studies may provide better quality evidence regarding these outcomes. We were also limited to outcomes where meta-analyses were performed. On account of these, we may have missed high quality observational studies. Another weakness is that we assessed papers based on the data published. Some AMSTAR criteria may have been fulfilled during the conduct of the study, but were excluded from the published manuscript resulting in lower quality scores. Finally, there are new work hour arrangements, such as flexible work hours or compressed workweeks, that we did not include in this review. Health effects of these arrangements might be similar to those included here if these work arrangements induce circadian rhythm disruptions or exceed the standard work hour length.

## Conclusion

Non-standard work hours are likely associated with several chronic outcomes. There is moderate grade evidence linking shift work to breast cancer and long work hours to stroke. There is low-grade evidence that suggests an increased risk of depression, some forms of cardiovascular diseases, and complications of pregnancy with exposure to non-standard work hours. Differential risk across different types of shift work and diverse populations should also be studied. Our results suggest that non-standard work hours may be detrimental to employee health. Workers should be informed of the potential risks associated with these jobs, and additional screening and preventive measures for breast cancer and stroke may be warranted. Higher quality research needs to be conducted to ascertain effects on other chronic and high-cost outcomes and to guide stronger policy recommendations.

## Supporting information

**S1 File. Search strategy.**
(PDF)

**S2 File. PRISMA checklist.**
(PDF)

**S3 File. List of excluded articles in full text screen.**
(XLSX)

**S4 File. Pooled results of included reviews per condition.**
(PDF)

**S5 File. Registered protocol.**
(PDF)

## Author Contributions

**Conceptualization:** Maxwell Akanbi, Linda C. O'Dwyer, Megan McHugh.

**Data curation:** Adovich S. Rivera, Maxwell Akanbi, Linda C. O'Dwyer, Megan McHugh.

**Formal analysis:** Adovich S. Rivera, Maxwell Akanbi, Megan McHugh.

**Funding acquisition:** Megan McHugh.

**Methodology:** Adovich S. Rivera, Linda C. O'Dwyer, Megan McHugh.

**Project administration:** Adovich S. Rivera.

**Supervision:** Megan McHugh.

**Visualization:** Adovich S. Rivera.

**Writing – original draft:** Adovich S. Rivera.

**Writing – review & editing:** Adovich S. Rivera, Maxwell Akanbi, Linda C. O'Dwyer, Megan McHugh.

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
