## [Decision Letter · Decision Letter 0]

15 Jan 2020

PONE-D-19-34056

Shift Work and Long Work Hours and their Association with Chronic Health Conditions: A Systematic Review of Systematic Reviews with Meta-analyses

PLOS ONE

Dear Mr Rivera,

Thank you for submitting your manuscript to PLOS ONE. After careful consideration, we feel that it has merit but does not fully meet PLOS ONE’s publication criteria as it currently stands. Therefore, we invite you to submit a revised version of the manuscript that addresses the points raised during the review process.

We would appreciate receiving your revised manuscript by Feb 29 2020 11:59PM. To enhance the reproducibility of your results, we recommend that if applicable you deposit your laboratory protocols in protocols.io, where a protocol can be assigned its own identifier (DOI) such that it can be cited independently in the future. For instructions see: http://journals.plos.org/plosone/s/submission-guidelines#loc-laboratory-protocols

We look forward to receiving your revised manuscript.

Kind regards,

Omid Beiki, M.D., Ph.D.

Academic Editor

PLOS ONE

Reviewers' comments:

Reviewer's Responses to Questions

**Comments to the Author**

1. Is the manuscript technically sound, and do the data support the conclusions?

Reviewer #1: Yes

2. Has the statistical analysis been performed appropriately and rigorously? 

Reviewer #1: Yes

3. Have the authors made all data underlying the findings in their manuscript fully available?

Reviewer #1: Yes

4. Is the manuscript presented in an intelligible fashion and written in standard English?

Reviewer #1: Yes

5. Review Comments to the Author

Reviewer #1: This is a very comprehensive review. I would have liked more conceptual discussion on potential underlying mechanisms. For example, I would have liked more on the link between work patterns and breast cancer. That work would affect breast cancer onset seems to me like a stretch.

Then there is an issue of heterogeneity of effects. I would expect that work-related stress would have a greater effect on the probability of stroke among 64-year olds than among 34 year olds. More on such heterogeneity would be desirable to have.

For some worker types, night shifts are unavoidable, e.g. health workers, firemen, police.

6. PLOS authors have the option to publish the peer review history of their article (what does this mean?). If published, this will include your full peer review and any attached files.

Reviewer #1: No

---

## [Author Response · Author response to Decision Letter 0]

27 Feb 2020

Detailed below are our responses to the comments. These can also be found in the Response to comments file we uploaded.

Comment 1: Please ensure that your manuscript meets PLOS ONE's style requirements, including those for file naming.*

Response: We have made the necessary changes to fulfill PLOS ONE’s style requirements. Since they were numerous and minor, we did not include these among the tracked changes. We hope this makes the manuscript more readable.

Comment 2: This is a very comprehensive review. I would have liked more conceptual discussion on potential underlying mechanisms. For example, I would have liked more on the link between work patterns and breast cancer. That work would affect breast cancer onset seems to me like a stretch.

Response: We agree with the comment regarding the conceptual discussion and have added more detailed examples of biological pathways that link shift work and long work hours with chronic conditions in both the introduction and discussion. To align with our results, we focused on mechanisms that explain the link between breast cancer and stroke to non-standard work hours. Changes made with corresponding page and line numbers are below:

• Page 3, Lines 55 to 60 New statements:

“For example, breast cancer among female shift works has been attributed to increasing DNA methylation with increasing exposure to shift work[4]. Long work hours, meanwhile, not only cuts into non-work hours that the body needs for rest and recovery, but can also be a form of psychologic stress [2,6], and if chronically exposed, this stress can lead to cardiovascular disease [7]. Non-standard work hours can also induce unhealthy coping behaviors, such as low physical activity and poor diets [2].”

• Page 28, lines 302 to 306

“We found studies that proposed disease mechanisms for two conditions with moderate grade evidence. For breast cancer, shift work leads to disruption of circadian rhythms which in turn lead to genetic and epigenetic changes that promote cancer growth [4]. For stroke, long work hours is a source of stress, and this stress leads to damage to the cardiovascular system. Long work hours can also promote unhealthy behaviors that further increase risk for stroke [6,7].”

Comment 3: Then there is an issue of heterogeneity of effects. I would expect that work-related stress would have a greater effect on the probability of stroke among 64-year olds than among 34 year olds. More on such heterogeneity would be desirable to have.

Response: We were also interested in any important subgroups such as age and sex. However, not all reviews included these in their analysis. We have added the results of the limited subgroup analyses related to stroke. We have previously reported subgroups based on shift type and total length of shift work exposure for breast cancer. We did not include additional subgroups for other conditions since the overall evidence for these conditions was low or very low in quality. We have included the new statements with corresponding page and line numbers below:

• Page 25, Lines 229 to 231

“Their metaregression results suggested that risk might be higher among those with high socio-economic status (compared to low socio-economic status) but there were no differences by age group or sex. “

• Page 29, Lines 312 to 318

“We downgraded much of the evidence due to issues of heterogeneity. The individual studies pooled by meta-analyses that we reviewed often had differences in definitions, measurement of exposures, and included populations. There were also differences in the variables used to calculate adjusted risk estimates. Despite these potential sources of heterogeneity, subgroup analyses often failed to identify any socio-demographic or study design-related factors as a significant source of heterogeneity. Fortunately, individual level cohort data is increasingly becoming available and allows for individual-level meta-analyses.”

Comment 4: For some worker types, night shifts are unavoidable, e.g. health workers, firemen, police.

Response: We agree with this comment and have added this detail in both the introduction and discussion.

• Page 3, Lines 45 to 46

“Some jobs such as those in healthcare, manufacturing, and law enforcement routinely require night time or prolonged shifts.”

• Page 23, Lines 282 to 283

“This policy should be considered for occupations where shift work is unavoidable, such as healthcare and law enforcement.”

---

## [Editor Report · Decision Letter 1]

16 Mar 2020

Shift Work and Long Work Hours and their Association with Chronic Health Conditions: A Systematic Review of Systematic Reviews with Meta-analyses

PONE-D-19-34056R1

Dear Dr. Rivera,

We are pleased to inform you that your manuscript has been judged scientifically suitable for publication and will be formally accepted for publication once it complies with all outstanding technical requirements.

With kind regards,

Omid Beiki, M.D., Ph.D.

Academic Editor

PLOS ONE
---

## [Editor Report · Acceptance letter]

18 Mar 2020

PONE-D-19-34056R1 

Shift Work and Long Work Hours and their Association with Chronic Health Conditions: A Systematic Review of Systematic Reviews with Meta-analyses 

Dear Dr. Rivera:

I am pleased to inform you that your manuscript has been deemed suitable for publication in PLOS ONE. Congratulations! Your manuscript is now with our production department. 

With kind regards,

on behalf of

Dr. Omid Beiki 

Academic Editor

PLOS ONE